# Impact of BMI on Survival Outcomes of Immunotherapy in Solid Tumors: A Systematic Review

**DOI:** 10.3390/ijms22052628

**Published:** 2021-03-05

**Authors:** Alice Indini, Erika Rijavec, Michele Ghidini, Gianluca Tomasello, Monica Cattaneo, Francesca Barbin, Claudia Bareggi, Barbara Galassi, Donatella Gambini, Francesco Grossi

**Affiliations:** Medical Oncology Unit, Fondazione IRCCS Ca’ Granda Ospedale Maggiore Policlinico, 20122 Milan, Italy; alice.indini@policlinico.mi.it (A.I.); erika.rijavec@policlinico.mi.it (E.R.); michele.ghidini@policlinico.mi.it (M.G.); gianluca.tomasello@policlinico.mi.it (G.T.); aprile83@gmail.com (M.C.); francesca.barbin@policlinico.mi.it (F.B.); claudia.bareggi@policlinico.mi.it (C.B.); barbara.galassi@policlinico.mi.it (B.G.); donatella.gambini@policlinico.mi.it (D.G.)

**Keywords:** obesity, cancer, immunotherapy, survival, irAEs, anti-PD1, anti-PDL1, anti-CTLA4, obesity paradox

## Abstract

Growing research has focused on obesity as a prognostic factor during therapy with immune-checkpoint inhibitors (ICIs). The role of body-mass index (BMI) in predicting response and toxicity to ICIs is not clear, as studies have shown inconsistent results and significant interpretation biases. We performed a systematic review to evaluate the relationship between BMI and survival outcomes during ICIs, with a side focus on the incidence of immune-related adverse events (irAEs). A total of 17 studies were included in this systematic review. Altogether, the current evidence does not support a clearly positive association of BMI with survival outcomes. Regarding toxicities, available studies confirm a superimposable rate of irAEs among obese and normal weight patients. Intrinsic limitations of the analyzed studies include the retrospective nature, the heterogeneity of patients’ cohorts, and differences in BMI categorization for obese patients across different studies. These factors might explain the heterogeneity of available results, and the subsequent absence of a well-established role of baseline BMI on the efficacy of ICIs among cancer patients. Further prospective studies are needed, in order to clarify the role of obesity in cancer patients treated with immunotherapy.

## 1. Introduction

Immunotherapy with immune-checkpoint inhibitors (ICIs) has demonstrated to provide survival benefit in a growing number of cancer patients. To date, antibodies targeting the programmed cell death 1 (PD-1), its ligand (PD-L1), and the cytotoxic T lymphocyte-associated antigen 4 (CTLA-4), are approved for the treatment of several solid tumors [1,2,3,4,5,6,7,8,9,10]. Despite the recognized efficacy of ICIs, still most patients receiving immunotherapy will experience treatment resistance (either primary, adaptive, or acquired), and eventually progress and die for cancer [11]. It is now known that ICIs′ resistance is a heterogeneous phenomenon, which comes as the result of a complex interplay among the host immune system, cancer cells, and tumor microenvironment [11,12]. Several components contribute to the onset of treatment resistance, and most of them are largely unknown.

Over the last years, growing research has focused on obesity as a factor that could predict patients′ response to immunotherapy [13]. Obesity is defined by high body mass index (BMI) (≥30 kg/m^2^) according to the World Health Organization (WHO) standard definition [14]. Data from several epidemiologic studies have demonstrated an inverse correlation between BMI and mortality due to cardiovascular disease and other chronic conditions, a phenomenon called the “obesity paradox” [15]. To date, the impact of obesity on cancer patients has not been clearly defined. Higher BMI represents a recognized risk factor for cancer, and it also correlates with worse outcome in several kinds of tumors [16]. However, obesity seems to act as a double-edged weapon in patients receiving immunotherapy. Excessive adiposity is associated with chronic low-grade inflammation [17] potentially improving survival outcomes during ICIs, an evidence supported by both preclinical and clinical data [13,18]. Leptin, one of the most important adipokine, contributes to the induction of an immunosuppressive microenvironment by increasing PD-1 expression, and pro-inflammatory cytokines such as tumor necrosis factor-α (TNF α) and interleukin 6 (IL-6), and by decreasing immune-stimulatory molecules [19]. Leptin receptors are expressed throughout the immune system cells and are involved in both innate and adaptive immune responses [19]. Thus, obesity could have favorable effects during treatment with ICIs. Preclinical studies have shown increased markers of T cell exhaustion in tumor-infiltrating lymphocytes from obese mice, leading to higher biologic aggressiveness and increased tumor progression [13]. However, this leptin-induced T cell exhaustion is reversed in mice tumors treated with anti-PD1, an evidence subsequently confirmed also in obese cancer patients [13]. Figure 1 shows the pathophysiological background of obesity, and the mediators involved in the immune responses that are also common targets of ICIs.

Several studies have tried to establish the role of BMI in predicting response or resistance to immunotherapy, however with inconsistent results and significant interpretation biases [20]. The aim of this systematic review is to present the current evidences on the relationship between BMI and survival outcomes during ICIs, with a side focus on the incidence of immune-related adverse events (irAEs).

## 2. Results

A total of 456 citations were retrieved upon systematic literature search, among which 198 were duplicates and 218 were initially excluded through reviewing titles and abstracts. The remaining 46 records were screened, and 14 were excluded as not reporting original data. The remaining 26 studies were subsequently reviewed and screened according to our inclusion and exclusion criteria. Finally, 18 studies were included in our systematic review (Figure 2) [18,21,22,23,24,25,26,27,28,29,30,31,32,33,34,35,36,37]. The included studies were published between 2018 and 2020, and were retrospective studies, either single or multicenter; one study was a pooled analysis of patients treated with immunotherapy in the context of phase 2 and 3 trials. Table 1 displays the main characteristics of the studies included in the analysis. Overall, most studies reported data of patients with melanoma (*n* = 5), non-small cell lung cancer (NSCLC) (*n* = 5), and renal cell carcinoma (RCC) (*n* = 3); indeed, these were the most common diagnoses even among patients included in the remaining 5 studies (see Table 1 for more details). Appendix A displays the search strategy used, and results of the search from each database. Table 2 displays the quality score according to the Newcastle Ottawa Scale (NOS), and the risk of biases assessment. Table 3 reports the statistical methods and results for each analyzed study.

### 2.1. Melanoma

Melanoma was far the most common diagnosis among patients included in the analyzed studies. Overall, five studies reported data on the correlation of BMI in melanoma patients receiving ICIs. Evidence from Richtig et al. showed that increased BMI had a positive correlation with disease response to ipilimumab (odds ratio (OR) 2.80 (95% confidence interval, CI 1.06 ± 7.39); *p* = 0.037), with a non-significant trend towards longer overall survival (OS) (HR 1.81 (95% CI 0.98–3.33), *p* = 0.056), and without correlation with progression free survival (PFS) (HR 1.03 (95% CI 0.62–1.70) [22]. Similarly, in a small population of Japanese patients, low BMI combined with high C-reactive protein to albumin ratio (CAR) correlated with early progressive disease during treatment with nivolumab (OR 0.048 (95% CI 0.0047–0.49); *p* = 0.011) [23]. The study by McQuade et al. is the only large multicenter study (*n* = 331) supporting a positive association of high BMI with improved outcomes, however only in male patients receiving anti-PD1 or PD-L1 (PFS HR 0.69 (95% CI 0.45–1.06), *p* = 0.07; OS HR 0.69 (95% CI 0.42–1.12); *p* = 0.84) [18]. Overall, two studies, making the largest sample size (total *n* = 704), reported no association of BMI with disease response and survival [21,24]. In the study by Rutkowski et al., BMI was not associated with disease control rate (DCR) (adjusted OR 0.98 (95% CI 0.93–1.03); *p* = 0.432), PFS (HR 1.00 (95% CI 0.98–1.03) *p* = 0.732), nor OS (HR 1.02 (95% CI 0.99–1.05); *p* = 0.202) [21]. Similarly, Young et al. reported no differences in DCR (OR 0.58 (95% CI 0.31–1.09); *p* = 0.09), PFS (HR 1.28 (95% CI 0.90–1.83); *p* = 0.18), and OS (HR 1.10 (95% CI 0.72–1.67); *p* = 0.65), according to BMI [24].

### 2.2. NSCLC

NSCLC was the second most common diagnosis among the analyzed studies, with four studies reporting data on NSCLC only patients. A pooled analysis of patients treated with atezolizumab in 4 international, multicenter clinical trials (the BIRCH, the FIR, the POPLAR, and the OAK trials), reported a significant improvement in OS for obese patients receiving immunotherapy (HR 0.64 (95% CI 0.51–0.81); *p* < 0.001) [37]. The association with survival improvement was even strongest among patients with high PD-L1 expression (HR 0.48 (95% CI 0.34–0.66); *p* < 0.001) [37]. The large multicenter study by Cortellini et al. on 962 PD-L1 positive (≥50%) NSCLC patients receiving pembrolizumab found a positive association of high BMI with objective response rate (ORR) (adjusted OR 1.61 (95% CI 1.04–2.50); *p* = 0.0208), PFS (adjusted HR 0.61 (95% CI 0.45–0.82); *p* = 0.0012), and OS (HR 0.70 (95% CI 0.49–0.99); *p* = 0.0474) [25]. In contrast, a Japanese study on 226 patients treated with anti-PD1 showed an association of BMI with ORR (HR 3.00 (95% CI 1.12–7.60); *p* = 0.0106), and disease control rate (DCR) (HR 2.45 (95% CI 1.26–4.76); *p* = 0.0075), but failed to demonstrate an association with survival (Table 2) [28]. It should be noted, however, that the cutoff for overweight category in this study was BMI ≥19.1 kg/m^2^ [28]. Then, two smaller single-center reports found no association of BMI with oncological outcomes, but showed that pre-treatment weight loss (HR 1.19, *p* = 0.16) [27] and low subcutaneous fat mass (SCFM) (HR 0.75, *p* = 0.006) [26] were predictors for poor survival outcomes in NSCLC patients treated with nivolumab.

### 2.3. RCC

Overall, three studies reported data on BMI among RCC patients receiving ICIs, and were all consistent in showing a positive association of BMI with survival outcomes [29,30,31]. The study by De Giorgi et al. evaluated patients treated with nivolumab in the Italian Expanded Access Program (EAP) (*n* = 313) [29]. In this study, higher BMI was not correlated with different ORR (*p* = 0.58), and DCR (*p* = 0.13), but correlated with better OS (HR 1.58 (95% CI 1.09–2.28); *p* = 0.01) [29]. One study evaluated BMI as a part of a scoring system which also considered monocyte-to-lymphocyte ratio (MLR), and the number and sites of metastases at baseline (the Emory scoring system), confirming that higher BMI had a positive association with survival outcomes [30]. The study by Labadie et al. found a significantly positive association of BMI with PFS (HR 0.87 (95% CI 0.79–0.96); *p* = 0.007), and a trend towards improved OS (HR 0.19 (95% CI 0.03–1.11); *p* = 0.07) among obese patients experiencing clinical benefit to treatment [31].

### 2.4. Other Solid Tumors

Considering the five studies reporting data on patients receiving ICIs for mixed solid tumors, the most common tumor diagnosis was melanoma, NSCLC, and RCC (for details on cancer diagnosis among different studies see Table 1). Overall, four studies reported a positive association of BMI with both response rates and survival outcomes [33,34,35,36]. In the study by Martini et al., data of patients with advanced solid tumors treated in phase 1 clinical trials showed that higher BMI (treated as a continuous variable) was associated with better PFS (HR 0.96 (95% CI 0.92–1.00); *p* = 0.03), and OS (HR 0.92 (95% CI 0.87–0.97); *p* = 0.001). The study by Cortellini et al. showed that overweight and obese patients had significantly longer time to treatment failure (TTF) (HR 0.51 (95% CI 0.44–0.60); *p* < 0.0001), PFS (HR 0.46 (95% CI 0.39–0.54); *p* < 0.0001), and OS (HR 0.33 (95% CI 0.28–0.41); *p* < 0.0001) [34]. The other study by Cortellini et al. confirmed a correlation of higher BMI with improved survival outcomes, especially among overweight patients (log-rank across all BMI subgroups: *p* < 0.0001) [35]. It is worthy to underline, though, that the data presented in the two multicenter Italian studies are presumably overlapping between the two reports, and should be evaluated accordingly [34,35]. The study by Rogado et al. confirmed that overweight patients had better ORR (*p* < 0.001), and PFS (*p* = 0.01) [36]. Only one study did not show a correlation of baseline BMI with survival outcomes, but showed that overweight patients had significantly better ORR (*p* = 0.04). Interestingly, this study also showed that patients with any pretreatment decrease in BMI had worse OS (HR 1.61 (95% CI 1.27–2.05); *p* < 0.001) [32].

### 2.5. Correlation of BMI with irAEs Incidence

Overall, seven studies evaluated the potential correlation of BMI with treatment-related toxicity, along with the correlation of BMI with survival outcomes. Among these, two studies on melanoma [18,24], two studies on NSCLC [26,37], and one study reporting data of patients with solid tumors [36] found no impact of BMI on the incidence of treatment-related toxicities. In the study by McQuade et al. the incidence of irAEs among patients treated with anti-PD1 and PD-L1 was superimposable regardless of BMI category (incidence of all grades irAEs: 31% for normal weight, 33% for overweight, and 36% for obese patients) [18]. Similarly, Young et al. found no impact of higher BMI on the incidence of treatment related toxicity (OR 0.80 (95% CI 0.44–1.46); *p* = 0.47) [26]. The study by Popinat et al., found a comparable incidence of irAEs (all grades) across patients′ subgroups; grade >1 toxicity was not statistically correlated to 1-year survival rate (*p* = 0.33) [26]. In the pooled analysis performed by Kichenadasse et al., no significant differences were seen in the frequency of irAEs across BMI categories (26% among normal weight, 29% among overweight, and 32% among obese patients; *p* = 0.73), except for skin-related irAEs (HR 1.47 (95% CI 1.2–2.0) for overweight patients [37]. The study by Rogado et al. found that the incidence of irAEs was not different in patients with excess weight (*p* = 0.21). However, the association of high BMI and irAEs was correlated with a marked prognostic trend in ORR (OR 161, *p* < 0.00001), and in PFS (HR 5.89; *p* < 0.001) [36]. In total, two multicenter studies from Cortellini et al. reported a higher incidence of irAEs among obese patients [34,35]. In the first study [34], overweight and obese patients were significantly more likely to experience irAEs (any grade) compared to normal weight patients (*p* < 0.0001), without significant difference in the incidence of serious (i.e., grades 3–4) irAEs (*p* = 0.1338). In the second study [35], both overweight and obesity were predictors for irAEs of any grade at both univariate and multivariate analysis. Obesity was the only factor significantly related to a higher incidence of grade 3–4 irAEs (OR 11.9 (95% CI 6.4–22.3); *p* < 0.0001) and irAEs leading to treatment discontinuation (OR 8.8 (95% CI 4.3–18.2); *p* < 0.0001) [35]. However, as mentioned before, these data should be interpreted considering a potential rate of overlap between the populations of these two studies.

## 3. Discussion

Immunotherapy has gained unprecedented success in oncology over the last years. Predictive biomarkers have been extensively studied to identify potential key factors for treatment resistance [38]. A consistent body of research has focused on the impact of patient-associated factors, such as sex, age, and BMI, on ICIs efficacy. In this systematic review, we evaluated the association between BMI and survival outcomes of cancer patients receiving ICIs, and we focused on the incidence of treatment-related adverse events. A thorough analysis of selected studies revealed that the heterogeneity of reported data made a quantitative analysis rather difficult to be performed. The heterogeneity of data was mainly due to differences in BMI categorization (e.g., dichotomic versus continuous variable; single variable versus part of a risk scoring system), but most importantly to different cutoffs for the definition of overweight and obese categories. Hence, we decided to carry out a descriptive analysis of studies according to the cancer type, assuming that this would make study populations more homogeneous in terms of patients and oncologic treatments’ characteristics. The current evidence is not enough to support a clearly positive association of BMI with survival outcomes. Regarding toxicities, available studies confirm a superimposable rate of irAEs among obese and normal weight patients.

Excess body fat is a complex metabolic disorder correlated with chronic systemic meta-inflammation, resulting in dysregulation of immune responses [39]. To date, little is understood about the real impact of obesity on immune responses and treatment resistance in cancer patients treated with ICIs. Overall, there seems to be a complex multifactorial relationship among the adipose tissue, the tumor and the host immune system [17]. Several factors are involved in this interplay, including sex, age, dietary differences, gut microbiome composition, and nutritional status [40,41,42]. As an example, the impact of obesity is sex-specific, as suggested by more favorable prognosis of female melanoma patients compared to males [40], as well as better survival outcomes in male patients treated with ICIs [18]. Sex-based immunological characteristics are due to differences in steroid hormone levels (namely estrogens), which influence the functional activity of innate immune cells and downstream adaptive immune responses [41]. Furthermore, gender and obesity also impact on the composition of gut microbiome with well-known implications on ICIs efficacy [42]. ICIs metabolism can be compromised in obese patients, due to the altered metabolism of free fatty acids and the presence of pro-inflammatory cytokines in the liver [43]. From a pharmacological point of view, the dosage of most ICIs monoclonal antibodies has been recently converted from a weight-based to a flat dose [44]. A potential alteration of ICIs clearance, together with the diffuse use of flat doses, and the lack of precise data on the pharmacokinetics of immunotherapy in obese patients, might have relevant implications during ICIs treatment in this setting. This characteristic might be exclusive to ICIs compared with other anticancer drugs. In fact, it has been widely demonstrated that chemotherapy should be used on a full weight-basis in the treatment of obese patients with cancer, particularly when the goal of treatment is cure [45]. Conversely, targeted therapy doses are independent from patients’ weight, although the real impact of BMI also on this class of drugs has not been fully understood yet [46].

The potential role of BMI on the incidence of irAEs is of concern. The onset of irAEs has been extensively correlated with increased responses and better survival outcomes during cancer immunotherapy [47]. It is therefore interesting to evaluate whether the presence of obesity is independent prognostic factor for irAEs incidence together with higher response. Even if ideally intriguing, only isolated reports have confirmed a possible association of toxicity with better outcomes of ICIs [34,35,36].

Overall, several factors are implicated in the relationship among cancer, obesity, response and toxicity of immunotherapy, making it challenging to explore this issue with the currently available evidences. Several selection and methodic biases should be carefully considered. First, BMI has been widely used as a surrogate for obesity, however it does not reflect more specific measures of body composition (i.e., skeletal muscle, lean mass, and adipose tissue) nor the distribution of adipose tissue (i.e., subcutaneous vs. visceral fat) [48]. Sarcopenic obesity (i.e., obesity with depleted muscle mass) is predictive of worse survival outcomes and toxicity in cancer patients receiving chemotherapy [49]. Body composition analyses, together with parameters of nutritional status, might more precisely define prognosis and response of cancer patients receiving ICIs [24].

Second, as other mechanisms of immunotherapy resistance, the role of adipose tissue is dynamic and changes over time. As such, obese patients might be more responsive to ICIs in the first phase of treatment, but with time become resistant due to progressive irreversible T cell exhaustion and consequent worse outcomes. Most of the available studies only provide static pretreatment BMI assessment and its correlation with survival. This leads to difficult interpretation of clinical results, which can vary over time according to the correlation of BMI with response and outcomes at different timepoints. Pretreatment BMI trends and dynamic on-treatment evaluation, rather than single baseline assessment, can more accurately reflect patients’ nutritional status and general conditions [21].

Finally, available studies share intrinsic limitations related to their retrospective nature, heterogeneity of patients’ cohorts (in terms of cancer diagnosis and ICIs treatments) and statistical methods, and differences in obesity definition, including inconsistency in BMI categorization for overweight and obese patients across different studies. Altogether, these factors might explain the heterogeneity of available results, and the subsequent absence of a well-established role of baseline BMI on the efficacy of ICIs among cancer patients.

## 4. Materials and Methods

The study search was designed to include population criteria, systemic treatment(s), and survival outcomes. Our systematic review was modeled according to the Preferred Reporting Items for Systematic Reviews and Meta-Analyses statement [50], and it was registered in the International Prospective Register of Systematic Reviews (available at http://www.crd.york.ac.uk/PROSPERO/#myprospero, accessed on 5 March 2021; CRD224807. Registration date: 6 December 2020). As such, PubMed, Scopus, Web of Science, EMBASE and Cochrane Central databases were systematically searched on 7 December 2020, using the terms “BMI”, or “obesity”, and “outcome”, and “immunotherapy” or “anti-CTLA4”, or “anti-PD1”, or “anti-PD-L1”, or “ipilimumab”, or “tremelimumab”, or “nivolumab”, or “pembrolizumab”, or “atezolizumab”, or “durvalumab”, or “avelumab”, in combination with “melanoma”, or “NSCLC”, or “urologic neoplasms”, or “renal cell carcinoma”, or “cancer”.

Using these search criteria, we identified all English language original reports, assessing the association between BMI and ICIs efficacy in patients with solid tumors. For this purpose, we included eligible studies according to the following inclusion criteria: (1) studies reporting data of patients diagnosed with solid tumors and treated with ICIs, either as a single agent or in combination; (2) studies reporting data on BMI, either categorized into groups according to cutoff values, or as a continuous variable; (3) studies reporting data on disease response (i.e., objective response rate [ORR], best overall response [BOR], and disease control rate [DCR]), and survival outcomes (i.e., progression-free [PFS], and overall survival [OS]). When overlapping data were reported in different studies, we included the most recent or highest quality study. References of the included articles were further searched to identify other potentially relevant studies. Exclusion criteria included duplicate publications, publications reporting overlapping data, non–English language literature, case reports, case series including less than 10 patients, abstracts, letters, editorials and commentaries, and reviews and metanalyses not reporting original data.

In each report, we sought to extract the following variables: BMI, diagnosis, oncologic treatment(s), and treatment outcomes. Data on oncologic treatment(s) included: type of ICIs (anti-PD1, anti-CTLA4), monotherapy or combination therapy. Oncologic outcomes included: disease response (i.e., ORR, BOR, and DCR), and survival (i.e., PFS, and OS). Data on irAEs frequency and its correlation with BMI were also recorded. Selected studies were assessed for quality and risk of bias, applying the Newcastle-Ottawa Scale (NOS).

## 5. Conclusions

There is a strong rationale for a correlation of excess adiposity with cancer prognosis. However, it is still unclear whether obesity can be considered a positive prognostic factor in cancer, rather than a condition of immune dysfunction that can be exploited by ICIs, resulting in heightened efficacy. The current evidences cannot support a straight causal relationship between obesity and outcomes of ICIs. Prospective studies are needed, in order to clarify the role of obesity in cancer patients treated with immunotherapy. The evaluation of parameters including anthropometric measures and laboratory analysis, will contribute to define the role of metabolism and nutritional status, with potential future therapeutic implications.

## Figures and Tables

**Figure 1 ijms-22-02628-f001:**
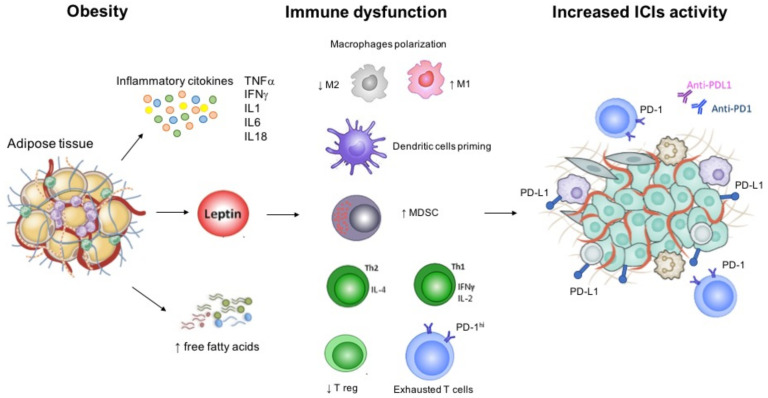
Immune dysfunction related with obesity and background for increased immunotherapy activity. Abbreviations: IFN, interferon; IL, interleukin; MDSC; myeloid derived stem cell; PD-1, programmed cell death 1; PD-L1, programmed cell death ligand 1; Th, T helper; TNF, tumor necrosis factor; T reg, regulatory T cell.

**Figure 2 ijms-22-02628-f002:**
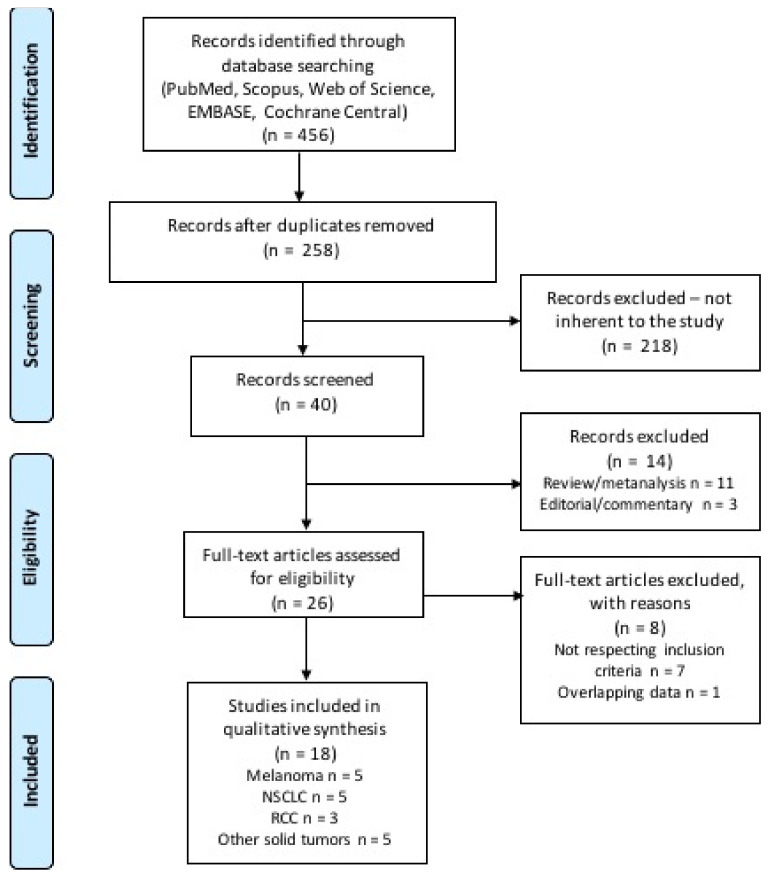
Flow diagram of literature research and study selection.

**Table 1 ijms-22-02628-t001:** Main characteristics of the included studies.

Author, Year	Type of Study	Country(ies)	Cancer Type(s)	Sample Size	ICI(s)	BMI Cutoffs	Median BMI	Obesity and/or Overweight Rate *	Survival Outcomes	irAEs	Comments
Kichenadasse, 2020	Pooled analysis of phase 2–3 trials	International	NSCLC	1434	atezolizumab	Overweight 25–29.9 kg/m^2^ Obese ≥30 kg/m^2^	N.A.	7%	OS, PFS	no difference in irAEs incidence according to BMI	High BMI associated with better OS. Association was consistent for men and women. Obese patients with high PD-L1 expression have best survival outcomes.
Rutkowski, 2020	Retrospective multicenter	Italy, Poland	melanoma	417	Ipilimumab nivolumab pembrolizumab	Overweight 25–29.9 kg/m^2^ Obese ≥30 kg/m^2^	26.3 (23.7–29.7)	23.3%	PFS, OS, DCR	N.A.	No association of BMI with DCR, PFS and OS. The interaction between BMI and gender was not statistically significant.
Johannet, 2020	Retrospective single center	U.S.	Solid tumors ^1^	629	anti-CTLA4 anti-PD1/PD-L1 anti-CTLA4 + anti-PD1	Normal <25 kg/m^2^ Overweight ≥25 kg/m^2^	N.A.	N.A.	BOR, ORR, DCR, PFS, OS	N.A.	No association of baseline BMI with response and survival. Decreasing pretreatment BMI associates with worse response and survival
Young, 2020	Retrospective single center	U.S.	melanoma	287	ipilimumab + nivolumab atezolizumab nivolumab pembrolizumab	Overweight 25–29.9 kg/m^2^ Obese ≥30 kg/m^2^	28.9 (16.7–50.6)	40.1%	PFS, OS, ORR	no differences in irAEs incidence (any grade) according to BMI	No association of BMI with ORR, PFS and OS. High TATI associated with decreased RR and PFS among women.
Takada, 2020	Retrospective single center	Japan	NSCLC	226	nivolumab pembrolizumab	Low <19.1 kg/m^2^ High ≥19.1 kg/m^2^	21.7 (13.9–36.2)	78.3%	PFS, OS, ORR, DCR	N.A.	Positive association of high BMI with ORR. No impact on PFS and OS.
Cortellini, 2020	Retrospective multicenter	Italy	NSCLC PD-L1 ≥50%	962	pembrolizumab	Overweight 25–29.9 kg/m^2^ Obese ≥30 kg/m^2^	24.2 (14.0–44.9)	12.2%	PFS, OS, ORR	N.A.	Positive association of high BMI with ORR, and prolonged PFS and OS
Cortellini, 2020	Retrospective multicenter	Italy	NSCLC melanoma RCC Other ^2^	1070	atezolizumab nivolumab pembrolizumab	Overweight 25–29.9 kg/m^2^ Obese ≥30 kg/m^2^	25 (13.6–46.6)	12.1%	PFS, OS	Higher incidence of irAEs in obese patients	Positive association of higher BMI with PFS and OS.
Rogado, 2020	Retrospective single center	Spain	Solid tumors ^3^	132	nivolumab pembrolizumab	Normal <25 kg/m^2^ Overweight ≥25 kg/m^2^	24.9 (14.8–37.1)	50%	ORR, PFS	no difference in irAEs incidence according to BMI	Positive association of high BMI with ORR and PFS.
Martini, 2019	Retrospective single center	U.S.	Solid tumors treated in phase I clinical trials ^4^	90	ICIs + experimental agents anti-PD-L1 experimental IO agent	Overweight 25–29.9 kg/m^2^ Obese ≥30 kg/m^2^	27.4 (14.9–45.6)	25.6%	PFS, OS	N.A.	Positive association of higher BMI with PFS and OS.
Magri, 2019	Retrospective single center	Israel	NSCLC	46	nivolumab	BMI analyzed as continuous variable	23.8 (14.9–39.3)	N.A.	OS	N.A.	No association of BMI with OS. Weight loss is a prognostic parameter.
Kondo, 2019	Retrospective single center	Japan	melanoma	39	nivolumab	Low <20 kg/m^2^ High ≥20 kg/m^2^	23 (15.0–35.9)	N.A.	PFS, OS, EPD	N.A.	Low BMI and high CAR are associated with EPD.
Popinat, 2019	Retrospective single center	France	NSCLC	55	nivolumab	Overweight 25–29.9 kg/m^2^ Obese ≥30 kg/m^2^	24.7 (18.0–34.1)	N.A.	ORR, OS	no difference in irAEs incidence according to BMI	No association of BMI with ORR and OS. Low SCFM is associated with poor OS.
Cortellini, 2019	Retrospective multicenter	Italy	NSCLC melanoma RCC Other ^5^	976	atezolizumab nivolumab pembrolizumab	Normal <25 kg/m^2^ Overweight ≥25 kg/m^2^	24.9 (13.5–46.6)	11%	ORR, TTF, PFS, OS	Higher incidence of irAEs (any grade) in obese patients	Positive correlation of BMI with ORR, TTF, PFS and OS. Better survival results among female patients
De Giorgi, 2019	Italian EAP	Italy	RCC	313	nivolumab	Normal <25 kg/m^2^ Overweight ≥25 kg/m^2^	N.A.	49.8%	ORR, OS	N.A.	No association of BMI with ORR. Positive association of BMI with OS.
Martini, 2019	Retrospective single center	U.S.	RCC	100	ipilimumab + nivolumab nivolumab	Normal <25 kg/m^2^ Overweight ≥25 kg/m^2^	26.7	60%	PFS, OS	N.A.	Positive association of BMI with survival (within the Emory scoring system)
Labadie, 2019	Retrospective multicenter	U.S., Canada, Spain	RCC	90	atezolizumab nivolumab pembrolizumab	Overweight: 25–29.9 kg/m^2^ Obese ≥30 kg/m^2^	N.A.	23%	PFS, OS	N.A.	Positive association of BMI with PFS and OS
McQuade, 2018	Retrospective multicenter	U.S.	melanoma	331	atezolizumab nivolumab pembrolizumab	Overweight: 25–29.9 kg/m^2^ Obese ≥30 kg/m^2^	N.A.	36%	PFS, OS	no differences in irAEs incidence according to BMI	Positive impact of BMI in male patients.
Richtig, 2018	Retrospective multicenter	Austria	melanoma	76	ipilimumab	Normal <25 kg/m^2^ Overweight ≥25 kg/m^2^	25.6 (18.0–59.1)	47%	PFS, OS, ORR	N.A.	Positive correlation with RR. No impact on PFS. Trend towards longer OS.

* definition of overweight and obese categories varies according to body mass index (BMI) cutoffs among different studies; in studies reporting both overweight and obese categories, only data regarding obese patients are reported. ^1^ Study population included: melanoma (*n* = 268), non-small cell lung cancer (NSCLC) (*n* = 128), renal cell carcinoma (RCC) (*n* = 37), breast (*n* = 36), head and neck (*n* = 25), liver (*n* = 25), genitourinary (*n* = 21), brain (*n* = 18), soft tissue (*n* = 18), non-melanoma skin (*n* = 14), gastric (*n* = 13), ovarian (*n* = 11), pancreatic (*n* = 9) or uterine cancer (*n* = 6). ^2^ Other tumors, unspecified (2.7%). ^3^ Study population included: NSLC (*n* = 93), melanoma (*n* = 12), head and neck carcinoma (*n* = 9), RCC and urothelial bladder carcinoma (*n* = 10), Hodgkin lymphoma (*n* = 3), gastric (*n* = 2), gallbladder (*n* = 1), Merkel cell (*n* = 1), hepatocellular carcinoma (*n* = 1). ^4^ Study population included: melanoma (*n* = 30), gastrointestinal tract tumors (*n* = 20), NSCLC or head and neck (*n* = 18), breast cancer (*n* = 11), other not specified (*n* = 11). ^5^ Other tumors, unspecified (2.4%). Abbreviations: BMI, body-mass index; BOR, best overall response; CAR, C-reactive protein to albumin ratio; CTLA-4, cytotoxic T-lymphocyte antigen 4; DCR, disease control rate; EAP, expanded access program; EPD, early progressive disease; ICIs, immune-checkpoint inhibitors; IO, immunooncology; irAEs, immune-related adverse events; N.A., not assessed; NSCLC, non-small cell lung cancer; ORR, objective response rate; OS, overall survival; PD-1, programmed cell death 1; PD-L1, programmed cell death ligand 1; PFS, progression free survival; RCC, renal cell carcinoma; RR, response rate; SCFM, subcutaneous fat mass; TATI, total adipose tissue index; TTF, time to treatment failure; U.S., United States.

**Table 2 ijms-22-02628-t002:** Quality and risk of bias assessment of the analyzed studies.

Author, Year	Quality Assessment (NOS)				Random Sequence Generation	Allocation Concealment	Blinding of Participants or Personnel	Blinding of Outcome Assessment	Incomplete Outcome Data	Reporting Bias/Selective Reporting	Other Sources of Bias Was the Study Apparently Free of Other Problems that Could Put It at High Risk of Bias?
	Selection	Comparability	Exposure/Outcome	Total Score							
Kichenadasse, 2020	***	**	***	8	(−)	(−)	(−)	(−)	(+/−)	(+)	(+/−)
Rutkowski, 2020	***	**	**	7	(−)	(−)	(−)	(−)	(−)	(+/−)	(+/−)
Johannet, 2020	***	*	**	6	(−)	(−)	(−)	(−)	(−)	(+/−)	(−)
Young, 2020	***	*	**	6	(−)	(−)	(−)	(−)	(−)	(+/−)	(−)
Takada, 2020	**	*	**	5	(−)	(−)	(−)	(−)	(−)	(+/−)	(−)
Cortellini, 2020	***	*	**	6	(−)	(−)	(−)	(−)	(−)	(+/−)	(−)
Cortellini, 2020	***	*	**	6	(−)	(−)	(−)	(−)	(−)	(+/−)	(−)
Rogado, 2020	***	*	**	6	(−)	(−)	(−)	(−)	(−)	(+/−)	(−)
Martini, 2019	***	*	***	8	(−)	(−)	(−)	(−)	(−)	(+/−)	(+/−)
Magri, 2019	**	*	**	5	(−)	(−)	(−)	(−)	(−)	(+/−)	(−)
Kondo, 2019	**	*	**	5	(−)	(−)	(−)	(−)	(−)	(+/−)	(−)
Popinat, 2019	**	*	**	5	(−)	(−)	(−)	(−)	(−)	(+/−)	(−)
Cortellini, 2019	***	*	**	6	(−)	(−)	(−)	(−)	(−)	(+/−)	(−)
De Giorgi, 2019	***	*	***	7	(−)	(−)	(−)	(−)	(−)	(+/−)	(−)
Martini, 2019	***	*	**	6	(−)	(−)	(−)	(−)	(−)	(+/−)	(−)
Labadie, 2019	***	*	**	6	(−)	(−)	(−)	(−)	(−)	(+/−)	(−)
McQuade, 2018	***	**	**	7	(−)	(−)	(−)	(−)	(−)	(+/−)	(−)
Richtig, 2018	**	*	**	5	(−)	(−)	(−)	(−)	(−)	(+/−)	(−)

Abbreviations: NOS, Newcastle Ottawa Scale. Legend: (−) = High risk of bias; (+) = Low risk of bias; (+/−) = Unclear.

**Table 3 ijms-22-02628-t003:** Overview of statistical methods and results of the analyzed studies.

Author, Year	Statistical Method(s)	Survival Outcome(s)	HR, 95% CI	*p* Value
Kichenadasse, 2020	Cox proportional hazards regression models	PFS	0.88 (0.78–0.99)	0.03
		OS	0.64 (0.51–0.81)	<0.001
Rutkowski, 2020	Cox proportional hazards regression models	PFS	1.00 (0.98–1.03)	0.732
		OS	1.02 (0.99–1.05)	0.202
Johannet, 2020	Cox proportional hazards regression models	PFS	1.01 (0.99–1.03)	0.33
		OS	0.99 (0.96–1.02)	0.38
Young, 2020	Log-rank test	PFS	1.28 (0.90–1.83)	0.18
		OS	1.10 (0.72–1.67)	0.65
Takada, 2020	Cox proportional hazards regression models	PFS	1.47 (1.04–2.05) *	0.0269 *
		OS	1.59 (1.10–2.30) *	0.0138 *
Cortellini, 2020	Cox proportional hazards regression models	PFS	0.61 (0.45–0.82)	0.0012
		OS	0.70 (0.49–0.99)	0.0474
Cortellini, 2020	Log-rank test	PFS	NA	<0.0001
		OS	NA	<0.0001
Rogado, 2020	Log-rank test	PFS	3.77 (1.33–10.66)	0.01
Martini, 2019	Cox proportional hazards regression models	PFS	0.96 (0.92–1.00)	0.03
		OS	0.92 (0.87–0.97)	0.001
Magri, 2019	Cox proportional hazards regression models	OS	1.19 (0.93–1.51)	0.16
Kondo, 2019	Cox proportional hazards regression models	PFS	4.12 (1.84–9.22)	*p* = 0.001
Popinat, 2019	Cox proportional hazards regression models	OS	0.84 (NA) *	0.007 *
Cortellini, 2019	Cox proportional hazards regression models	PFS	0.46 (0.39–0.54)	<0.0001
		OS	0.33 (0.28–0.41)	<0.0001
De Giorgi, 2019	Cox proportional hazards regression models	OS	1.58 (1.09–2.28)	0.01
Martini, 2019	Cox proportional hazards regression models	OS	NA ^1^	NA ^1^
Labadie, 2019	Cox proportional hazards regression models	PFS	0.87 (0.79–0.96)	0.007
		OS	0.19 (0.03–1.11)	0.07
McQuade, 2018	Cox proportional hazards regression models	PFS	0.63 (0.41–0.95)	0.07
		OS	0.54 (0.34–0.86)	0.84
Richtig, 2018	Log-rank test	PFS	1.03 (0.62–1.70)	0.924
		OS	1.81 (0.98–3.33)	0.056

Abbreviations: CI, confidence interval; HR, hazard ratio; NA, not available; OS, overall survival; PFS, progression free survival. * Univariate analysis. ^1^ BMI evaluated as a part of the Emory scoring system.

## Data Availability

No new data were created or analyzed in this study. Data sharing is not applicable to this article.

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
