# Peer review of "Impact of BMI on Survival Outcomes of Immunotherapy in Solid Tumors: A Systematic Review"

_ijms, 2021, doi:10.3390/ijms22052628_

Round 1
Reviewer 1 Report
The authors present a systematic review on the use of immunotherapy across a broad spectrum of cancer diagnoses in relation to BMI/obesity. In my view, the manuscript is of interest to the clinical community and summarizes well valuable information; therefore, I believe it would be of interest to the readership of IJMS. However, there are some flaws that need to be addressed. The manuscript would benefit from further revision and clarification with respect to its limitations and the pathophysiological mechanisms behind the study hypothesis.
Reviewer´s comments.
- Please consider providing a figure on the potential targets of ICIs that are upregulated in obese potients and could explain better the pathophysiological background behind the study hypothesis, i.e. why this systematic review was undertaken. Please elaborate on these mechanisms in the introduction and discussion section.
- Please specify the databases that were searched through in the Prisma flow chart of the study (Fig 1). Cochrane Central, Embase and Medline/Pubmed ar the three databases that all systematic reviews should include, therefore please consider to include also Cochrane central.
- Please provide the systematic search strategy for the aforementioned databases in a supplementary file with the results of the search from each database.
- In Materials and Methods, please specify if there was a time limitation in the search strategy.
- There might be eligible studies not designed to assess differences in BMI in relation to the outcomes of interest. Therefore the terms BMI/obesity in the search strategy should not be included as a mesh term or a term in the title/abstract section only, as this would lead to a narrow search strategy, potentially missing valuable evidence.
- Please provide quality/risk for bias assessment of the included studies, applying the Newcastle-Ottawa Scale (NOS) or other validated assessment tool. A systematic review is as good as the included studies, therefore a quality assessment is essential according to Cochrane guidelines.
- Malnutrition associated with sarcopenia and low BMI is often encountered in advanced stages of different cancer diagnoses. Therefore the Authors should probably determine not only the role of ICIs in obese cancer patients, but also in patients with low BMI due to advanced disease stage. Please address this point by scrutinizing studies reporting outcomes of cancer patients with low BMI.
- With regards to data reported from the included studies in the results section, paragraphs 2.1-2.5 and Table 1, please comply to PRISMA guidelines. Hazard ratios, 95%CIs and p-values from e.g. log-rank survival analyses of the included studies should be reported thoughout. This applies also to the correlation analysis of BMI in relation to toxicities.
- Page 2-3, line 78-86. This part does not fit in the Results section and should be included in the Discussion instead.
Author Response
Comment: Please consider providing a figure on the potential targets of ICIs that are upregulated in obese patients and could explain better the pathophysiological background behind the study hypothesis, i.e. why this systematic review was undertaken. Please elaborate on these mechanisms in the introduction and discussion section.
Response: In order to comply with the reviewer’s comment, we added Figure 1 to our manuscript. Figure 1 1 shows the pathophysiological background of obesity, and the mediators involved in the immune responses that are also common targets of ICIs. We discussed the suggested topics in the Introduction section (Lines 47-63) and in the Discussion section (Lines 416-434).
Comment: Please specify the databases that were searched through in the Prisma flow chart of the study (Fig 1). Cochrane Central, Embase and Medline/Pubmed are the three databases that all systematic reviews should include, therefore please consider to include also Cochrane central.
Response: In order to comply with the reviewer’s observation, we specified the databases we searched through in the Prisma flow chart, and modified Figure 2 accordingly. We also included Cochrane central to the databases that were searched through.
Comment: Please provide the systematic search strategy for the aforementioned databases in a supplementary file with the results of the search from each database.
Response: In order to comply with the reviewer’s comment, we added Supplementary Table 1 reporting details of the results of the search from each database.
Comment: In Materials and Methods, please specify if there was a time limitation in the search strategy.
Response: In order to comply with the reviewer’s request, we specified the time in which the systematic search was performed (Line 448).
Comment: There might be eligible studies not designed to assess differences in BMI in relation to the outcomes of interest. Therefore, the terms BMI/obesity in the search strategy should not be included as a mesh term or a term in the title/abstract section only, as this would lead to a narrow search strategy, potentially missing valuable evidence.
Response: We agree with the reviewer that our search strategy can miss eligible studies not designed to assess differences in BMI in relation to the outcomes of interest. However, we specifically chose to maintain BMI and/or obesity as a mesh term or a term in the title/abstract section, in order to use a more selective search strategy. However, references of the included articles were thoroughly searched to identify other potentially relevant studies, and this led to a further improvement of articles selection with the identification of a higher number of eligible studies.
Comment: Please provide quality/risk for bias assessment of the included studies, applying the Newcastle-Ottawa Scale (NOS) or other validated assessment tool. A systematic review is as good as the included studies, therefore a quality assessment is essential according to Cochrane guidelines.
Response: In order to comply with the reviewer’s request, we added the quality/risk for bias assessment of the included studies, applying the Newcastle-Ottawa Scale (NOS) in Table 1. We also specified this in the Materials and Methods section (Lines 476-477).
Comment: Malnutrition associated with sarcopenia and low BMI is often encountered in advanced stages of different cancer diagnoses. Therefore, the Authors should probably determine not only the role of ICIs in obese cancer patients, but also in patients with low BMI due to advanced disease stage. Please address this point by scrutinizing studies reporting outcomes of cancer patients with low BMI.
Response: We thank the reviewer for this important observation. All the reported studies do not discriminate among patients with normal and/or low BMI, thus not allowing us to address the role of low BMI on survival outcomes. The rationale of these studies was to assess whether obesity, and not malnutrition or sarcopenia, was associated with outcomes and/or toxicity of immunotherapy. As mentioned in the Discussion section (Lines 419-425), BMI has been used as a surrogate for obesity, however it does not reflect more specific body composition nor the individual’s nutritional status. Indeed, in the Discussion section we discussed the role of malnutrition in association with obesity (i.e. sarcopenic obesity), as a potential predictive factor of worse survival outcomes and toxicity in cancer patients receiving chemotherapy. The potential role of low BMI, malnutrition and sarcopenia due to advanced disease would require a dedicated analysis and is beyond the aims of our research.
Comment: With regards to data reported from the included studies in the results section, paragraphs 2.1-2.5 and Table 1, please comply to PRISMA guidelines. Hazard ratios, 95%CIs and p-values from e.g. log-rank survival analyses of the included studies should be reported throughout. This applies also to the correlation analysis of BMI in relation to toxicities.
Response: In order to comply with the reviewer’s request, we added data on the statistical methods and results over the text (Results section), and we specifically added Table 2 reporting data on all the included studies.
Comment: Page 2-3, line 78-86. This part does not fit in the Results section and should be included in the Discussion instead.
Response: We thank the reviewer for this suggestion. We have moved this section to the Discussion section, accordingly (Lines 363-371).
Reviewer 2 Report
In this work authors provide a systematic review on the impact of excess BMI on patient outcome upon treatment with immunotherapies.
I congratulate the authors on their manuscript, which is straightforward and well-written (few typos to correct only). Methodology for conducting the review is well described. Authors identify several issues with the available studies which merit further investigation / clarification. The different settings (even different definitions for obesity) employed make difficult to draw proper conclusions on this topic. I found the article nice to read.
I have only minor suggestions:
Authors could also discuss in the Results section a bit more on further differences in studies regarding male/female ratio, since this is also related to distinct types of obesity, as they acknowledge in the discussion. This could even be added to Table 1.
Authors could also add a bit of discussion about the statistical methods used by different studies to assess associations between BMI and certain events, or even to assess impact on survival. This may represent an additional level of heterogeneity among studies.
For matter of scientific writing: authors mention several times “absence of association” or “an association between variable x and y”. Better say if there is a significant association or not significant, and if it is a positive or negative association.
Some typos:
Line 77: “diagnoses”.
Line 92: “by far”.
Line 178: Error Bookmark not defined, perhaps an error in reference.
Author Response
Reviewer #2.
Comment: Authors could also discuss in the Results section a bit more on further differences in studies regarding male/female ratio, since this is also related to distinct types of obesity, as they acknowledge in the discussion. This could even be added to Table 1.
Response: In order to comply with the reviewer’s observation, we provided data on sex related differences in the analyzed studies in Table 1, when reported.
Comment: Authors could also add a bit of discussion about the statistical methods used by different studies to assess associations between BMI and certain events, or even to assess impact on survival. This may represent an additional level of heterogeneity among studies.
Response: In order to comply with the reviewer’s request, we added Table 2 reporting details on the statistical methods used in the analyzed studies. We also added statistical heterogeneity as a limitation of included studies in the Discussion section (Line 436-437).
Comment: For matter of scientific writing: authors mention several times “absence of association” or “an association between variable x and y”. Better say if there is a significant association or not significant, and if it is a positive or negative association.
Response: We thank the reviewer for this important observation. We specified the presence of positive and/or significant association of BMI with either responses and survival outcomes throughout the manuscript and also in table 1.
Comment: Some typos: Line 77: “diagnoses”; Line 92: “by far”; Line 178: Error Bookmark not defined, perhaps an error in reference.
Response: In order to comply with the reviewer’s request, we corrected the suggested typos.
Round 2
Reviewer 1 Report
The Authors have answered adequately to my comments and have improved the quality of the manuscript.
I have some minor comments:
Table 1: please specify what is high BMI for each study. Is there a cut-off?
Table 2: I would suggest to modify this table, which should be restricted only to risk of bias assessment. Please remove the columns of statistical analyses from this table and provide this information where necessary in table 1 instead. Please provide a NOS star template providing the score for each of the following categories: selection, comparability, exposure/outcome.
Supplementary Table 1. Please provide the search strategy and results for each database. Mesh terms/search terms differ in each database. This is important as the Authors' methodology/search strategy should be reproducible.
Author Response
Reviewer #1.
Comment: Table 1: please specify what is high BMI for each study. Is there a cut-off?
Response: Regarding the reviewer’s comment, BMI cutoffs are provided in Table 1, column 7 for each of the analyzed study. Cutoffs for overweight and obese categories were provided when both cutoffs were analyzed in the study. Differences in results according to overweight and/or obese categories as compared with normal BMI are detailed in the text.
Comment: Table 2: I would suggest to modify this table, which should be restricted only to risk of bias assessment. Please remove the columns of statistical analyses from this table and provide this information where necessary in table 1 instead. Please provide a NOS star template providing the score for each of the following categories: selection, comparability, exposure/outcome.
Response: In order to comply with the reviewer’s request, we modified Table 2 and moved statistical methods and results to a dedicated table (Table 3). We added the NOS star template in Table 2, as requested.
Comment: Supplementary Table 1. Please provide the search strategy and results for each database. Mesh terms/search terms differ in each database. This is important as the Authors' methodology/search strategy should be reproducible.
Response: We thank the reviewer for this important observation. In order to comply with this request, we added the information regarding the search strategy in Supplementary Table 1.